# Is the extreme within-population genome size variation real in *Spodoptera frugiperda*?

**Karine Durand**, **Kiwoong Nam** *

DGIMI, INRAE, University of Montpellier, Montpellier, France

* ki-woong.nam@inrae.fr

## Abstract

Genome size variation is one of the main topics in evolutionary genomics. Recent studies in *Spodoptera frugiperda* (Insect; Lepidoptera) reported very striking results, including large genome size differences, up to a twofold variation within populations, with the existence of 1.37 Gb of non-reference genome sequences, which is 2.5 times larger than the reference genome, 544 Mb in size. These reports raise the question of whether such extreme genome size variations within populations can be biologically realistic. To evaluate these results, we analyzed reference genome assemblies and resequencing datasets from multiple independent studies, including those from the original research. We observed that high-quality reference genomes consistently range from 380 Mb to 390 Mb, aligning closely with flow cytometry measurements. The genome size estimates using k-mer-based approaches from field-collected samples across four independent studies suggest that extensive genome size variation within *S. frugiperda* is unlikely to occur. Additionally, genome size appears to have remained stable for at least 15.89 million years in the ancestral lineage of *S. frugiperda*. Taken together, these results do not support the existence of extreme genome size variation in *S. frugiperda*, emphasizing the need for careful validation.

## Introduction

Genome size variation has long been a key topic in evolutionary genomics [1], as the C-value paradox, the lack of correlation between genome size and organismal complexity, was coined as early as 1971 [2]. Genome size can vary substantially between species [3] or even between populations in a single species [4], primarily due to the differential accumulation of transposable elements [5]. Most copy number variation (or presence-or-absence variation) exists as rare alleles within populations, likely due to the deleterious effects [6–14]. Consequently, genome size variations are generally minor within populations. For example, in humans, nearly 30% of the 3.1 Gb genome can be subject to copy number variations [15], but a pangenomics study found that only 81 Mb of copy number variation sequences are shared among multiple individuals [16].

**Data availability statement:** This study was conducted using publicly available whole genome sequences from NCBI Genome (GCA_023101765.3, GCA_019297735.2, GCA_015832365.1, GCA_026413635.1, GCF_011064685.2, and GCA_900240015.1), NCBI SRA (ERR6937806, ERR6942234, SRR5132437, SRR27871596, PRJNA640063, PRJNA639295, PRJNA577869, PRJNA494340, and PRJNA639296), China National GeneBank Database (CNP0001020, CNA0003276), and Agricultural Genomics Institute at Shenzhen server (ftp://ftp.agis. org.cn/Spodoptera_Frugiperda/). Project data are available at: https://figshare. com/s/63497c4e2a06d6e8c891.

**Funding:** The study is supported by the department of Santé des Plantes et Environnement at Institut national de recherche pour l'agriculture, l'alimentation et l'environnement (Resistome). The funders had no role in study design, data collection and analysis, decision to publish, or preparation of the manuscript. We are grateful to Nicolas Nègre and Emmanuelle d'Alençon for their support in this study through intensive discussions.

**Competing interests:** The authors have declared that no competing interests exist.

However, recent studies suggest that *Spodoptera frugiperda* (fall armyworm; Insecta; Lepidoptera; Noctuidae) defies this trend. *S. frugiperda* is a phytophagous lepidopteran species feeding on diverse crops, including maize, rice, and sorghum, due to its extreme polyphagy. *S. frugiperda* is found on all continents except Antarctica [17], after the invasion from native North and South Americas in 2016 [18], with the involvement of a substantial lag phase in a non-native area [19]. Gui *et al.* [20] estimated genome sizes using a k-mer-based method from 32 individuals collected from China, Ethiopia, and the mainland USA. They reported a striking result that genome size varies, ranging from 510 Mb to 977 Mb, without detectable differences among sampling locations, implying extensive within-population genome size variations. They also generated a reference 544Mb genome assembly using 100 bp MGI-Seq.

Huang *et al.* [21] performed a follow-up pangenomics study using the same dataset and further investigated the cause of this surprising result. Their method is simple. First, 100 bp BGI-Seq reads from the field-collected samples of Gui *et al.* were mapped to the reference genome generated by Gui *et al.*, and unmapped reads were collected. Second, genome assembly was performed from these reads, and the resulting assemblies were considered to be non-reference genome sequences. Surprisingly, the resulting non-reference sequences totaled 1.37 Gb, which is nearly 2.5 times larger than the reference genome itself and far exceeds the known extreme case in the Mediterranean mussel, where non-reference genomes comprise nearly 48% of the reference genome (1.28 Gb and 580 Mb for reference and non-reference genomes, respectively) [22]. Thus, in *S. frugiperda*, genome sizes could be distributed between 510 Mb and 977 Mb by subsetting 26.6% to 51.0% of the total available sequence (1,914 Mb), including 544 Mb from the reference genome and 1,370 Mb from the non-reference assembly. If validated, this result would represent a milestone in evolutionary genomics, suggesting unprecedented within-population genome plasticity.

We identified three reasons that necessitate the validation of their findings. First, the assumption that all unmapped reads represent non-reference genomes needs further justification. If a genetic position contains a high level of heterozygosity, mapping of reads can be challenging. Then, a large proportion of unmapped reads might represent highly heterozygous loci rather than non-reference genomes. Second, the size of the used reference genome assembly should be robustly evaluated. Gui *et al.*'s reference genome assembly is 544 Mb in size, as mentioned earlier, which is significantly larger than the genome size measured by flow cytometry (396 Mb ± 3 Mb) from a laboratory corn strain colony originally seeded in Guadeloupe [23] using the standardized protocol of Johnston *et al.* [24]. Additionally, the reported genome sizes significantly exceed those of the NCBI reference genome generated using Oxford Nanopore long reads (384 Mb, NCBI accession number: GCA_023101765) and other published assemblies using PacBio long reads (380–390 Mb [25–28]), calling into question the accuracy of genome size exceeding 500 Mb. Third, it is difficult to imagine how homologous recombination can readily occur between a pair of haplotype genomes with two-fold size differences. A population with highly heterogeneous

genome sizes is likely to experience fitness depression, ultimately resulting in a constrained range of efficiently recombinable genome sizes within a population.

Motivated by the principle that striking scientific observations require rigorous validation, we conducted a multi-pronged approach to assess the reported genome size variation by Gui *et al.* For this purpose, we analyzed the reference genome sequences of Gui *et al.* [20], other public reference genome assemblies, and the non-reference genome sequence of Huang *et al.* [21] to test whether the claimed genome size variation (510–997 Mb) is realistic. We also used k-mer-based methods with a resequencing dataset from 163 samples, which include the 32 samples reported to have extensive genome size variation by Gui *et al.*, in addition to three other independent, publicly available datasets, including 302 samples across Argentina, Benin, Brazil, China, French Guiana, Guadeloupe, India, Kenya, Mexico, Puerto Rico, and the mainland USA, to evaluate whether similar levels of genome size variation are observed in other datasets. Finally, to place these results in an evolutionary context, we compared genome sizes across other *Spodoptera* species, providing insights into genome size dynamics within the genus.

## Result and discussion

### Reference and non-reference genome sequences

First, we assessed the accuracy of *S. frugiperda* reference genome assemblies according to the sizes using BUSCO genes [29], which are expected to exist predominantly as single-copy genes in a genome. We analyzed seven assemblies deposited in NCBI, along with the assembly generated by Gui *et al.* [20] (Table 1). The reference genome assembly from Gui *et al.* was the largest among these assemblies. When the number of Complete Single-copy BUSCO genes was below 5,020 (95% of the total 5,286 Lepidoptera BUSCO genes), assembly sizes varied between 329 Mb and 544 Mb (Fig 1A). However, when the number of Complete Single-copy BUSCO genes exceeded 5,020, assembly sizes converged to a range of 380–390 Mb, which closely aligns with the flow cytometry measurement (396 MB ± 3 MB). These genome assemblies were generated out of the samples collected from diverse locations, including Australia, Guadeloupe, the mainland USA, and Zambia, excluding the possibility of sampling bias.

Assemblies larger than 400 Mb contained significantly higher numbers of Complete Duplicated-copy BUSCO genes than the other assemblies (Fig 1B). We tested the possibility that the large size of Gui *et al.*'s genome assembly was due to natural segmental duplications. A total of 446 out of 462 BUSCO genes were found to be specifically duplicated in this assembly compared with the NCBI reference genome assembly (GCA_023101765.3; S1 Table). These duplicated genes were distributed across all 31 chromosomes of the NCBI reference genome. Such a pattern would only be consistent with whole-genome doubling occurring in the sample used for Gui *et al.*'s assembly. However, this explanation is unlikely, as

**Table 1. The analyzed genome assemblies and statistics for the BUSCO analysis.**

| NCBI Accession number | Location | Assembly size (bp) | Sequence number | Longest sequence length (bp) | N50 (bp) | Number of Complete and Single-copy BUSCO | Reference |
|---|---|---|---|---|---|---|---|
| CNA0003276* | China | 543,659,128 | 21,840 | 27,910,074 | 14,162,803 | 4,685 | Gui *et al.* (2022) [20] |
| GCA_023101765.3 | Australia | 383,907,753 | 68 | 22,093,432 | 13,003,764 | 5,189 | unpublished |
| GCA_012979215.2 | Zambia | 390,396,834 | 666 | 21,916,676 | 12,966,683 | 5,163 | Zhang *et al.* (2020) [28] |
| GCA_019297735.2 | Guadeloupe | 385,041,943 | 492 | 21,042,509 | 13,567,047 | 5,111 | Fiteni *et al.* (2022) [27] |
| GCA_015832365.1 | USA | 379,902,270 | 1,054 | 7,849,854 | 1,129,192 | 5,071 | Nam *et al.* (2020) [25] |
| GCA_026413635.1 | Kenya | 329,351,232 | 2,510 | 3,613,530 | 534,293 | 4,641 | unpublished |
| GCF_011064685.2 | China | 485,954,603 | 85 | 22,364,714 | 16,346,893 | 4,271 | unpublished |
| GCA_900240015.1 | Guadeloupe | 436,269,483 | 41,562 | 943,242 | 52,317 | 4,134 | Gouin *et al.* (2017) [23] |

*China National GeneBank Database.

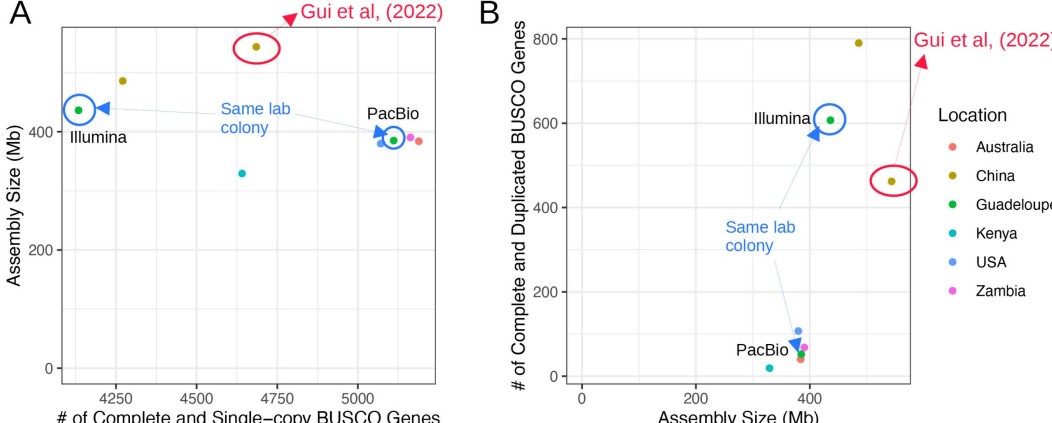

**Fig 1. The comparison of reference genome assemblies.** Relationships (A) between assembly size and the number of Complete and Single-Copy BUSCO genes and (B) between assembly size and the number of Complete and Duplicated BUSCO genes. The reference genome assembly generated by Gui *et al.* [20] is marked with red circles. Assemblies generated from a laboratory colony originating from Guadeloupe, sequenced using different technologies, are shown with blue circles and arrows.

only 8.43% (446/5286) of BUSCO genes were duplicated in this assembly. These results indicate that the precise genome size of *S. frugiperda* is close to 380–390 Mb. Consequently, the reference genome assembly by Gui *et al.* does not appear to reflect true genome size. This genome assembly could have been inflated during the assembly of reads into contigs by treating heterozygous alleles as non-allelic sites or, more likely, during scaffolding errors of the contigs using Hi-C.

It is worth noting that this result included two genome assemblies derived from the same laboratory colony, seeded from a population in Guadeloupe, but generated using different sequencing techniques. The assembly based on 150 bp Illumina short reads [23] was larger (436 Mb) and contained a higher number of duplicated BUSCO genes (607) than the assembly generated using PacBio long reads [26] (385 Mb and 52). This result suggests, again, that the observed assembly size variation does not reflect true genome size differences but is primarily driven by differences in the used techniques.

We tested the possibility that the non-reference genome sequences reported by Huang *et al.* actually contain the reference genome sequences of Gut *et al.* In total, 347 lepidopteran BUSCO genes were identified within the non-reference genome sequences. The vast majority of these genes (323/347 = 93.08%) were also present in the reference genome (S2 Table), indicating the dual presence of these genes in both datasets. Among these, 84 genes showed BLASTp hits with at least 50% coverage against the metazoan BUSCO genes, which are reasonably expected to exist as single-copy genes across animal species. These include genes encoding, for example, Wntless, DNA-directed RNA polymerase II subunit RPB11, and the NEDD8-activating enzyme E1 regulatory subunit.

Among the total of 8,603 protein-coding genes identified in the non-reference genome sequences, 918 genes had protein sequences that were 100% identical to those found in the reference genome (S1 Fig, S3 Table for the gene list). These results demonstrate that the non-reference genome sequences reported by Huang *et al.* actually include the reference genome sequence, suggesting that the 1.37 Gb non-reference genome size is overestimated.

### Genome size estimates from population data

We then re-evaluated the genome size variations (510 Mb – 997 Mb) reported by Gui *et al.* using GCE software [30], as they did. The resequencing dataset they generated includes 163 samples from the Americas, China, and Africa, among which 32 samples were used to estimate the genome size. The estimated genome sizes varied widely, ranging from

27.6 Mb to 1060.8 Mb (Fig 2A, left), implying 38.4-fold genome size variation (1060.8 Mb/27.6 Mb), which is biologically unrealistic. Fig 2A also shows that the genome size estimates did not appear to be influenced by sequencing throughput.

When we used Genomescope [31], another software using the reference-genome-free k-mer approach, the estimated range was reduced to 277.8 Mb – 446.7 Mb, with 80% of the samples falling within 323.3 Mb – 368.13 Mb (Fig 2A, left). Since a slight underestimation of genome size is expected when k-mer approaches are used [32], we believe this range does not significantly deviate from the flow cytometry measurements (396 MB ± 3 MB) and the size of the high-quality reference genome assemblies (380 Mb – 390 Mb). Thus, we conclude that Genomescope estimated a more realistic range of true genome size than GCE and that a two-fold genome size difference within populations is unlikely to be accurate or, at the very least, should be viewed with caution. According to Table S4 in the paper of Gui *et al.*, the median proportion of unmapped reads is only 3.09% across the samples. Such a small proportion is also unlikely to account for the twofold difference in genome size, as argued by Gui *et al.* [20]

Intriguingly, the estimated genome sizes are positively correlated with the lengths of both unique and repeat sequences within the genomes (S2 Fig). To compare the variation in genome size estimates between Genomescope and Gui *et al.*, we calculated the ratio of genome sizes from two randomly chosen samples with 10,000 replications. The average pairwise difference in genome sizes based on Genomescope was 7.23% (95% confidence interval: 0.16%–28.0%), which was much lower than the one based on the estimates of Gui *et al.* (25.7%, 95% confidence interval: 0.19%–70.5%). Notably, the distribution of differences from Gui *et al.*'s estimates was bimodal, a pattern not observed in the Genomescope results (Fig 2A, right). This result suggests that genome size estimation is highly sensitive to the used k-mer-based methods, with GCE potentially overestimating genome sizes in certain samples.

The resequencing dataset from Gui *et al.* primarily consists of *S. frugiperda* samples from China (20/32 = 62.5%). To further test extensive genome size variations in Chinese *S. frugiperda*, we analyzed an independently generated dataset from Zhang *et al.* [28], which includes 103 samples from China, using Genomescope. Once again, estimated genome sizes ranged from 295.4 Mb to 498.2 Mb, with 80% of the samples falling between 349.4 Mb and 387.4 Mb, regardless of

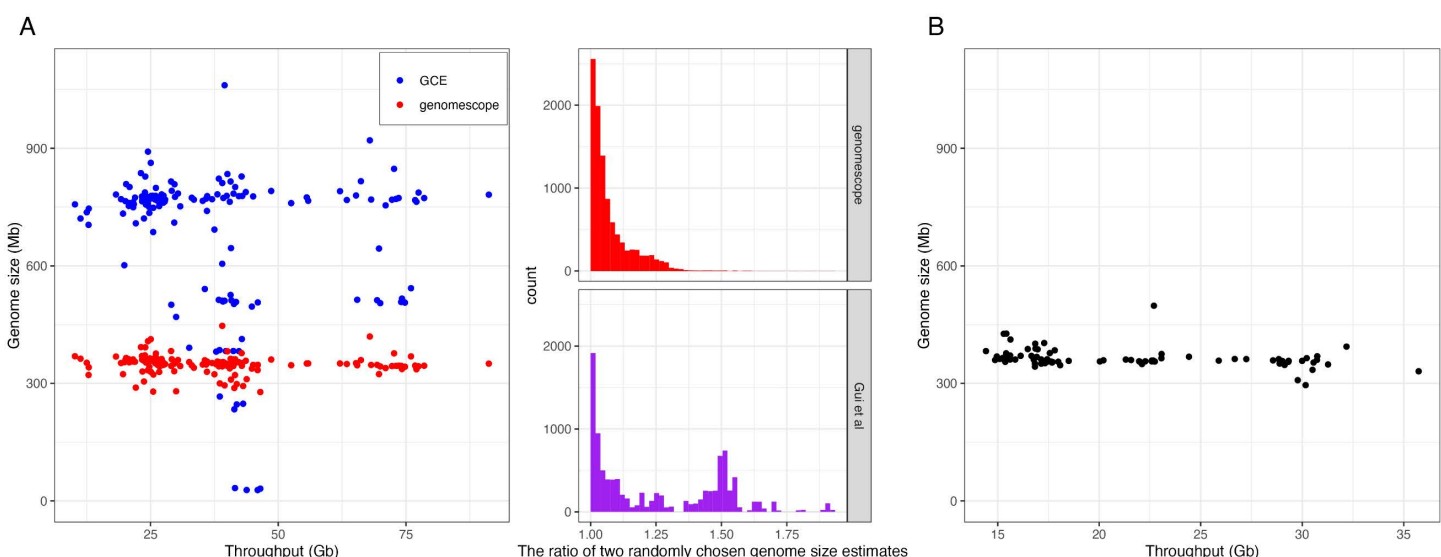

**Fig 2. K-mer-based genome size estimation from Chinese populations. (A)** Left: Estimated genome sizes from the resequencing dataset of Gui *et al.* [20] using GCE (blue dots) and Genomescope (red dots). Right: Distribution of ratios between two randomly selected genome size estimates from either Genomescope or Gui *et al.*, with 10,000 replications. **(B)** Estimated genome sizes from the dataset of Zhang *et al.* [28] using Genomescope, plotted against sequencing throughput for each sample.

sequencing throughput (Fig 2B). This result is inconsistent with the extensive genome size variation (510 Mb to 977 Mb) reported by Gui *et al.*

Additionally, we analyzed two independently generated datasets. The first dataset [33] included 55 samples collected from diverse geographic locations, including Argentina, Brazil, Kenya, Puerto Rico, and the mainland USA. Genomescope reported genome size estimation from 35 samples out of 55, all of which fell within the range of 349.9 Mb – 406.6 Mb, with a single exception reporting a genome size of 623.2 Mb (Fig 3A, left). This exceptional sample had the lowest model fit (Fig 3A, right), suggesting that this genome size estimate may not be biologically accurate.

The second dataset [34] comprised 144 samples, also collected from a wide geographic range including Benin, China, French Guiana, Guadeloupe, India, Mexico, Puerto Rico, and the mainland USA. Genomescope successfully reported the genome sizes from 133 samples. These samples had genome sizes between 325.6 Mb and 421.6 Mb for all samples, except for three outliers with the lowest model fit (Fig 3B). These results, again, do not support the argument by Gui *et al.* [20] of extensive genome size variation, exceeding 500 Mb in size.

### Genome size evolution in *Spodoptera*

To place the genome size variation of *S. frugiperda* in an evolutionary context, we inferred genome sizes for other *Spodoptera* species using publicly available sequencing data analyzed with Genomescope. These species included *S. exigua*, *S. picta*, *S. littoralis*, and *S. litura*, which diverged from the ancestral lineage leading to the extant *S. frugiperda* between 10.97 million years ago (95% confidence interval: 10.36–12.67 Mya) and 15.89 million years ago (95% confidence interval: 15.08–16.34 Mya), based on molecular clock estimates calculated from mitochondrial divergence calibrated with fossil data [35]. All of these species had genome sizes close to 400 Mb (Fig 4), suggesting that genome size has remained relatively stable in the ancestral lineage of *S. frugiperda*, at least over the past 15.89 million years.

### Conclusion

Taken together, these results do not support the claim that *S. frugiperda* exhibits extensive genome size variation between 510 Mb and 977 Mb [20] or the existence of 1.37 Gb of non-reference genome sequences in addition to a 544 Mb reference genome [21]. Instead, we demonstrate that the genome sizes of field-collected *S. frugiperda* samples remain largely close to the flow cytometry estimates (396 MB ± 3 MB). Furthermore, no dramatic changes in genome size have occurred

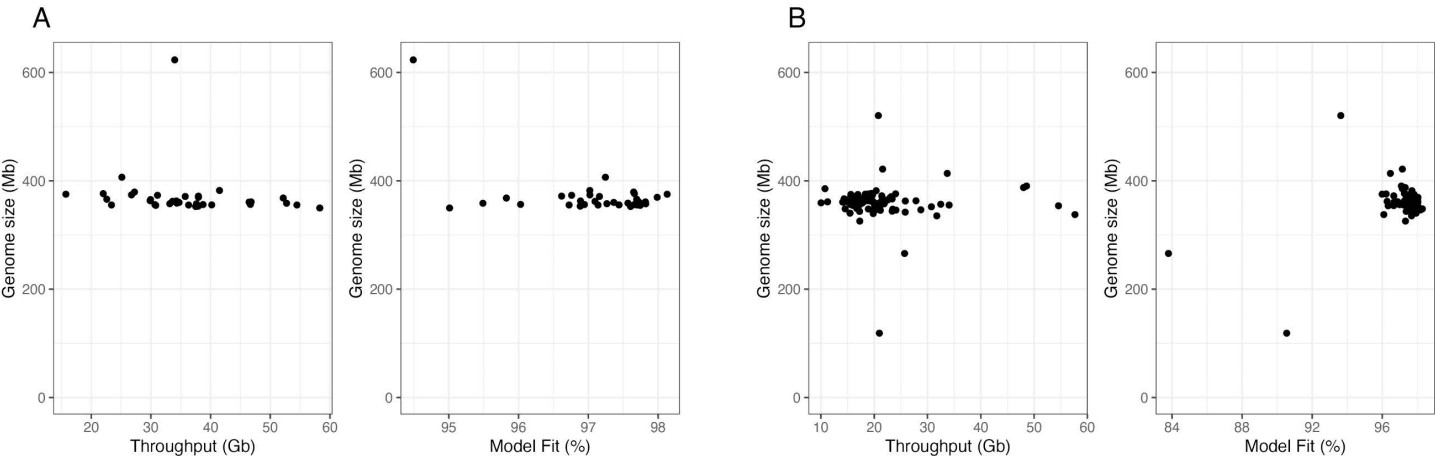

**Fig 3. Estimated genome sizes from the resequencing dataset of (A) Schlum *et al.* [33] and (B) Yainna *et al.* [34] using Genomescope according to the sequencing throughput for each sample or model fit.**

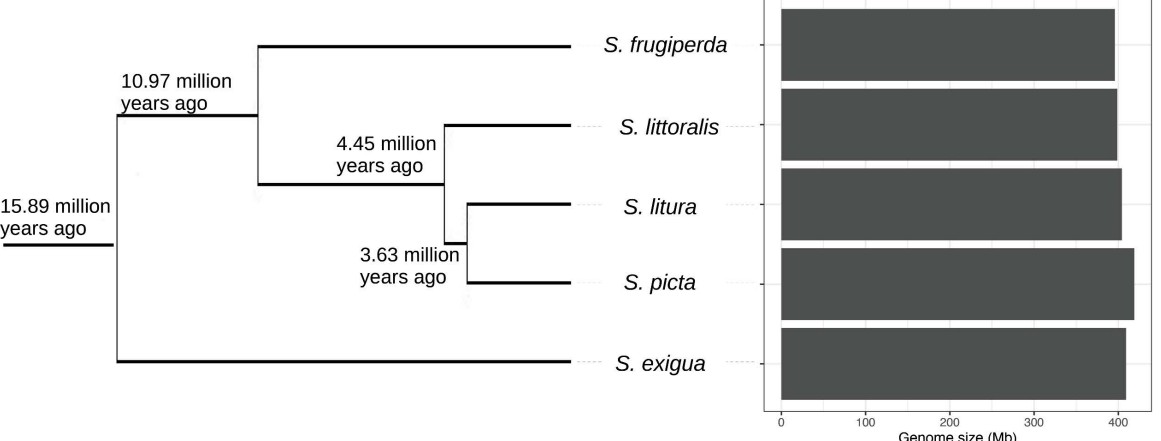

**Fig 4. Estimated genome sizes of *S. littoralis*, *S. litura*, *S. picta*, and *S. exigua* using Genomescope.** Phylogenetic relationships and divergence times were obtained from Kergoat *et al.* [35], and the genome size of *S. frugiperda* was based on flow cytometry measurements [23].

for at least 15.89 million years in the ancestral lineage of extant *S. frugiperda* insects. Therefore, we suggest that the results of Gui *et al.* and Huang *et al.* need to be interpreted with caution, and further validation is necessary before accepting the existence of extensive within-population genome size variation, such as a 2.5-fold larger non-reference genome compared to the reference genome.

Since the first *S. frugiperda* genome project in 2017 [23], at least four high-quality reference genomes have been published [25–28], all reporting genome sizes substantially smaller than those estimated by Gui *et al.*, as noted earlier. However, Huang *et al.* based their conclusions solely on the result of Gui *et al.* without addressing the discrepancies with prior research, effectively leaving the task of evaluating the pangenomic results to external researchers. We appreciate the value of open scientific discourse and view careful consideration of existing literature as a vital part of the research process.

## Methods

Reference genome assemblies listed in Table 1 were downloaded from NCBI Genomes or the China National GeneBank Database. BUSCO v5.2.2 analysis [29] was conducted with lepidoptera_odb10. Non-reference genome sequences were compared to reference genome sequences using the blastP of BLAST+ v2.12.0 [36]. Assembly statistics were calculated using the FastA.N50.pl script of the enveomics collection [37]. The resequencing reads from Gui *et al.* [20] were downloaded from the China National GeneBank Database (ID: CNP0001020). The resequencing reads from Zhang *et al.* [28] were downloaded from the Agricultural Genomics Institute at Shenzhen server (ftp://ftp.agis.org.cn/Spodoptera_Frugiperda/), generated from samples that were all collected from maize, with the exception of two samples collected from sugarcane. The resequencing reads from Schlum *et al.* [33] were obtained from NCBI SRA (ID: PRJNA640063). The dataset from Yainna *et al.* [34] deposited in NCBI SRA (ID: PRJNA639295, PRJNA577869, PRJNA494340, and PRJNA639296) originally included 177 samples. However, we used only 144 samples, as the remaining data was not publicly released in the original study [38]. The resequencing reads from *S. exigua*, *S. littoralis*, *S. litura*, and *S. picta* were downloaded from NCBI SRA (IDs: ERR6937806, ERR6942234, SRR5132437, and SRR27871596, respectively). Adapter sequences were removed from the reads using AdapterRemoval [39]. The distribution of k-mers (= 17) was calculated using Jellyfish v2.3.1 [40] from the filtered reads, and the genome size was estimated using Genomescope v1.0 [31]. The length of unique and repeat sequences for each assembly was also obtained from the result of Genomescope.

Alternatively, the distribution of k-mers (= 17) was calculated using kmerfreq v4.0 [30], and the genome size was estimated using Genomic Charactor Estimator (GCE) v1.0 [30].

## Supporting information

**S1 Table. Chromosomal locations on the NCBI reference genome assembly of BUSCO genes specifically duplicated in the genome assembly of Gui *et al* [20].**
(DOCX)

**S2 Table. The BUSCO genes found in the Reference genome assembly generated by Gui *et al.* or non-reference sequences generated by Huang *et al*.**
(DOCX)

**S3 Table. The genes showing 100% protein sequence identity between reference genome assemblies and non-reference sequences.**
(DOCX)

**S1 Fig. Histogram showing the number of genes from the non-reference genome sequences generated by Huang *et al.* [21] that were mapped to each gene with 100% identical protein sequences in the reference genome assembly generated by Gui *et al.* [20].**
(TIF)

**S2 Fig. The relationship between genome size and the lengths of repeat and unique sequences, as estimated by GenomeScope.**
(TIF)

## Acknowledgments

We are grateful to Nicolas Nègre and Emmanuelle d'Alençon for their support in this study through intensive discussions.

## Author contributions

**Conceptualization:** Karine Durand, Kiwoong Nam.

**Data curation:** Karine Durand.

**Formal analysis:** Kiwoong Nam.

**Writing – original draft:** Kiwoong Nam.

**Writing – review & editing:** Karine Durand, Kiwoong Nam.

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
