## [Decision Letter · Decision Letter 0]

8 Jun 2025

PONE-D-25-25103Is extreme within-population genome size variation real in Spodoptera frugiperda?PLOS ONE

Dear Dr. NAM,

Thank you for submitting your manuscript to PLOS ONE. After careful consideration, we feel that it has merit but does not fully meet PLOS ONE’s publication criteria as it currently stands. Therefore, we invite you to submit a revised version of the manuscript that addresses the points raised during the review process.

We look forward to receiving your revised manuscript.

Kind regards,

Vivekanandhan Perumal, Ph.D

Academic Editor

PLOS ONE

Journal Requirements:

“The study is supported by the department of Santé des Plantes et Environnement at Institut national de recherche pour l'agriculture, l'alimentation et l'environnement (Resistome). “

3. Please expand the acronym “INRAE” (as indicated in your financial disclosure) so that it states the name of your funders in full.

“The study is supported by the department of Santé des Plantes et Environnement at Institut national de recherche pour l'agriculture, l'alimentation et l'environnement (Resistome). We are grateful to Nicolas Nègre and Emmanuelle d'Alençon for their support in this study through intensive discussions.”

“The study is supported by the department of Santé des Plantes et Environnement at Institut national de recherche pour l'agriculture, l'alimentation et l'environnement (Resistome). “

5. Please ensure that you refer to Figure 1 in your text as, if accepted, production will need this reference to link the reader to the figure.

6. Please include your tables as part of your main manuscript and remove the individual files. Please note that supplementary tables (should remain/ be uploaded) as separate "supporting information" files

**Additional Editor Comments:**

After carefully reviewing this manuscript, I can see some merit in the research. However, in its current form, the manuscript is not scientifically sound. It contains several grammatical and typographical errors. I recommend a thorough revision of the entire manuscript, taking into account the comments from both reviewers.

Reviewers' comments:

Reviewer's Responses to Questions

**Comments to the Author**

1. Is the manuscript technically sound, and do the data support the conclusions?

Reviewer #1: Yes

Reviewer #2: Partly

2. Has the statistical analysis been performed appropriately and rigorously? 

Reviewer #1: Yes

Reviewer #2: N/A

3. Have the authors made all data underlying the findings in their manuscript fully available?

Reviewer #1: Yes

Reviewer #2: No

4. Is the manuscript presented in an intelligible fashion and written in standard English?

Reviewer #1: Yes

Reviewer #2: Yes

5. Review Comments to the Author

Reviewer #1: 1.Spelling format problem.

In the abstract: ' aligning breeze with flow cytometry measurement ', the results section: ' flow cytometry measurements ', it is recommended to unify the spelling.

In Figure 4: ' 10.97 mya ', ' 15.89 mya ', it is recommended to indicate the full name, such as ' 10.97 million years ago (mya) '.

The table and the figure note number are confused. The article mentions ' According to Table S4 of Gui et al. ', but the table number in the article is ' Table 1 ', and the supplementary material is ' Table S1 ', and ' S4 ' does not appear. When quoting the supplementary material form, it is necessary to confirm whether the number is correct.

The format of software names is not standardized. ' GCE software [29] recommends that the full name should be indicated when the software name first appears (such as Genome Complexity Estimator (GCE)).

2.In this paper, the BUSCO integrity of seven reference genomes was compared, but the specific influence mechanism of different assembly strategies on genome size was not clearly explained. It is recommended to supplement the comparison of assembly parameters of different technical routes to more clearly demonstrate the reason for the overestimation of the reference genome by Gui et al.

3.The representativeness of geographical population samples needs to be refined.

When analyzing Zhang et al. ' s Chinese population data, it is not clear whether the sample covers groups of different host plants (such as maize, rice). The host adaptability of Spodoptera frugiperda may affect the genome structure, and it is recommended to supplement the correlation analysis between host type and genome size.

4.The time node of evolutionary stability analysis needs to be determined.

The differentiation time of related species mentioned in this paper is 10.97-15.89 million years ago, but the calculation method of this time node is not explained. If there is uncertainty in the estimation of divergence time, it may affect the conclusion of genome stability, and the confidence interval of phylogenetic analysis needs to be supplemented.

5.Citation details of flow cytometry data are missing.

The results of flow cytometry ( 396 MB ± 3 MB ) were cited many times in this paper, but the specific source of the data ( such as sample collection location and experiment repetition times ) was not clear. It is recommended to supplement the experimental design in the original literature to enhance the credibility of the data.

6.The transparency of software version and data analysis process is insufficient.

The research methods mentioned the use of Genomescope and GCE, but did not specify the specific version number (such as Genomescope v2.0 vs v1.0). The difference between different versions of the algorithm may affect the results. It is necessary to supplement the software version information and expose the complete analysis script in the code library.

7.The verification method of non-reference sequence contamination can be increased.

The study only proved that the non-reference sequence contained reference genome contamination by BUSCO gene overlap (93 %) and protein sequence homology, but did not provide specific cases (such as which gene families had duplication). It is recommended to select representative genes (such as resistance-related genes) to verify whether they really belong to non-reference regions. Increase the rationality of the article.

Reviewer #2: In this article, Durand and Nam study genome size variation among the fall army worm Spodoptera frugiperda. This article should be considered as a response to two previous papers -Gui et al 2022 and Huang et al. 2025 that have reported significant genome size variation among this species. The authors have re-analyzed the data using :

1) BUSCO completion of the genome assemblies

2) Estimation of the genome size estimation of the re-sequencing data using k-mer distribution

Durand and Nam finally concluded that extensive genome size variation are unlikely and that the genome size in this species is stable, close to 400Mb and that larger (or smaller) genome assemblies reported so far are artifactual.

My general feeling is that the proofs provided by the authors are questionable and do not really support their central claim that genome size variation among this species never occurs. I believed that the data provided by the authors remain compatible with some substantial genome size variations among population/individuals. Here are my comments :

1) First of all, I have to mention that genome size variation is a very minor aspect of the work of Gui et al. 2022, it’s necessary to consult the supplementary data of the paper and only a short paragraph discusses why the genome assembly size reported in this study (543 Mb) is bigger than those reported so far (329 Mb to 400 Mb). Even previous assemblies resulting from the same sequencing technologies lead to substantial genome size variation (Table 1 and Figure 1). For example, PacBio assemblies vary from 320 Mb to 390 Mb which is huge. I agree with the authors that larger assemblies correlate with a higher number of duplicated BUSCO genes. But this would also be easily explained by natural segmental duplications that are not artifactual and do not result from assembly errors. At any moment the authors provide an explanation to understand the nature of the assembly artifacts that have led to bigger genome assemblies. My feeling is that natural variation based on genome duplications remains plausible and compatible with the BUSCO data reported here. I suggest the authors to carry some new analyzes to understand the origin of the extra 150Mb that are present in the Gui et al. 2022 assembly : numbers of extra contigs, size, read coverage, composition (transposons, duplicated genes...) for example. The goal is to identify some peculiar properties of these extra contigs that would be useful to decipher if there are “real” contigs or assembly artifacts.

2) Even the genome size estimation using k-mer lead to substantial variations that are systematically minored by the authors. For instance, in Fig 2A, genome size estimation with GenomeScope vary from 290 Mb to 450 Mb. Even if we excluded some extreme outliers, there is a significant spread within the 300Mb-400Mb range. Same patterns occurs in Fig. 3 with some huge genome size estimation variations. I agree with the authors that, in average, most samples fell close to the range of 400Mb but for reasons I ignore a fraction of the sample clearly deviate from the distribution and the authors failed to explain why. Maybe it’s artifactual in relation with sequencing bias or something like this, but again, it might reflect true genome size variation... Again, additional analyzes would be useful to understand why. I suggest the authors to try to assemble the read of these atypical samples and to analyzes the composition of the contigs, especially transposable elements that appears to be the main candidates for genome expansion. This would bring some valuable information to exclude the possibility that there are some degrees of natural genome variation.

3) I fully agree with the authors that reference genome assemblies of Spodoptera species are all close to 400 Mb (Fig 4) but if we enlarge the scope, there is huge genome size variation in Lepidoptera with some of them far exceeding the 500 Mb range (up to 800 Mb). The argument seems to me very week because it critically depends on the the phylogenetic range that were examined.

Minor points :

L48-57 / L168-176 : I think there is a misunderstanding of the pan-genome concept. The fact that there is a pangenome of 1.37 Gb found in 135 individuals that are not present in the reference genome assembly (400Mb) do not mean that a single individuals may have a genome of 400 Mb + 1.37 Gb = 1.77 Gb as mentioned by the authors. This simply mean that the 1.37 Gb of extra DNA may be randomly distributed among the 135 accessions. These 2 paragraphs need to be re-written.

L181-183 : The last sentence is offensive and usefulness. Remove it or replace it by a more neutral conclusion.

Figure 1 : Size and resolution of this figure need some improvements.

Data availability : it should be valuable to depose the raw data of the BUSCO and kmer distribution analyzes somewhere (Zenodo or something like this)

6. PLOS authors have the option to publish the peer review history of their article (what does this mean? ). If published, this will include your full peer review and any attached files.

**Do you want your identity to be public for this peer review?** For information about this choice, including consent withdrawal, please see our Privacy Policy .

Reviewer #1: **Yes: ** shaoying Wu

Reviewer #2: **Yes: ** Jonathan Filée

---

## [Author Response · Author response to Decision Letter 1]

14 Jun 2025

Reviewer #1:

R1: Thank you very much for the review. We did our best to improve the manuscript by addressing your comments.

 1.Spelling format problem.

In the abstract: ' aligning breeze with flow cytometry measurement ', the results section: ' flow cytometry measurements ', it is recommended to unify the spelling.

In Figure 4: ' 10.97 mya ', ' 15.89 mya ', it is recommended to indicate the full name, such as ' 10.97 million years ago (mya) '.

The table and the figure note number are confused. The article mentions ' According to Table S4 of Gui et al. ', but the table number in the article is ' Table 1 ', and the supplementary material is ' Table S1 ', and ' S4 ' does not appear. When quoting the supplementary material form, it is necessary to confirm whether the number is correct.

The format of software names is not standardized. ' GCE software [29] recommends that the full name should be indicated when the software name first appears (such as Genome Complexity Estimator (GCE)).

R1-1: Thank you for this comment. The figure was modified as you suggested. And we confirm that the table numbers are correct, including Table S4 of Gui et al. We also revised the manuscript to standardize the software name format (L227).

2.In this paper, the BUSCO integrity of seven reference genomes was compared, but the specific influence mechanism of different assembly strategies on genome size was not clearly explained. It is recommended to supplement the comparison of assembly parameters of different technical routes to more clearly demonstrate the reason for the overestimation of the reference genome by Gui et al.

R1-2: Thank you for this comment.

We agree that identifying the cause of the inflated genome size in Gui et al.’s assembly is an important issue. However, the analyzed assemblies include three that have not been described in published papers, making it impossible to find how they were generated. Furthermore, Gui et al. did not release scripts or intermediate files, which makes pinpointing the exact cause of the genome inflation very challenging.

Nevertheless, we believe that addressing this issue falls beyond the scope of the present study. A formal benchmarking study would be more appropriate for systematically comparing the performance of different sequencing technologies and assemblers.

We added assembly statistics in Table 1 and lines 211–212. We believe this information will help readers better evaluate the quality of the assemblies.

3. The representativeness of geographical population samples needs to be refined.

When analyzing Zhang et al. ' s Chinese population data, it is not clear whether the sample covers groups of different host plants (such as maize, rice). The host adaptability of Spodoptera frugiperda may affect the genome structure, and it is recommended to supplement the correlation analysis between host type and genome size.

R1-3: Thank you for this comment. All samples of Zhang et al. were collected from maize fields, with only two exceptions (GLZ1 and GLZ3 from sugarcane). We inserted this information (L213-L126).

4.The time node of evolutionary stability analysis needs to be determined.

The differentiation time of related species mentioned in this paper is 10.97-15.89 million years ago, but the calculation method of this time node is not explained. If there is uncertainty in the estimation of divergence time, it may affect the conclusion of genome stability, and the confidence interval of phylogenetic analysis needs to be supplemented.

R1-4: Thank you for this comment. We inserted detailed methodological information and confidence intervals (L183-L187).

5.Citation details of flow cytometry data are missing.

The results of flow cytometry ( 396 MB ± 3 MB ) were cited many times in this paper, but the specific source of the data ( such as sample collection location and experiment repetition times ) was not clear. It is recommended to supplement the experimental design in the original literature to enhance the credibility of the data.

R1-5: Thank you for this comment. We inserted detailed information obtained from the original article (L63-L65).

6.The transparency of software version and data analysis process is insufficient.

The research methods mentioned the use of Genomescope and GCE, but did not specify the specific version number (such as Genomescope v2.0 vs v1.0). The difference between different versions of the algorithm may affect the results. It is necessary to supplement the software version information and expose the complete analysis script in the code library.

R1-6: Thank you for this comment. We used Genomescope v1.0. We inserted this information (L224-L225).

7.The verification method of non-reference sequence contamination can be increased.

The study only proved that the non-reference sequence contained reference genome contamination by BUSCO gene overlap (93 %) and protein sequence homology, but did not provide specific cases (such as which gene families had duplication). It is recommended to select representative genes (such as resistance-related genes) to verify whether they really belong to non-reference regions. Increase the rationality of the article.

R1-7: Thank you for your comment. We agree that presenting specific examples would be beneficial.

We presented cases of 100% protein sequence identity between reference and non-reference genomes in Table S3 (and L128). We believe this list provides strong evidence that the genes in the reference genome assembly are also present in the non-reference assemblies. Additionally, as you suggested, we highlighted specific examples of metazoan BUSCO genes that are present in both reference and non-reference genome assemblies (L123–L126 and L129).

Reviewer #2: In this article, Durand and Nam study genome size variation among the fall army worm Spodoptera frugiperda. This article should be considered as a response to two previous papers -Gui et al 2022 and Huang et al. 2025 that have reported significant genome size variation among this species. The authors have re-analyzed the data using :

1) BUSCO completion of the genome assemblies

2) Estimation of the genome size estimation of the re-sequencing data using k-mer distribution

Durand and Nam finally concluded that extensive genome size variation are unlikely and that the genome size in this species is stable, close to 400Mb and that larger (or smaller) genome assemblies reported so far are artifactual.

My general feeling is that the proofs provided by the authors are questionable and do not really support their central claim that genome size variation among this species never occurs. I believed that the data provided by the authors remain compatible with some substantial genome size variations among population/individuals.

R2: Thank you for the review. We appreciate that your comments are constructive. We feel that our manuscript has been improved by addressing your comments.

Here are my comments :

1) First of all, I have to mention that genome size variation is a very minor aspect of the work of Gui et al. 2022, it’s necessary to consult the supplementary data of the paper and only a short paragraph discusses why the genome assembly size reported in this study (543 Mb) is bigger than those reported so far (329 Mb to 400 Mb). Even previous assemblies resulting from the same sequencing technologies lead to substantial genome size variation (Table 1 and Figure 1). For example, PacBio assemblies vary from 320 Mb to 390 Mb which is huge. I agree with the authors that larger assemblies correlate with a higher number of duplicated BUSCO genes. But this would also be easily explained by natural segmental duplications that are not artifactual and do not result from assembly errors. At any moment the authors provide an explanation to understand the nature of the assembly artifacts that have led to bigger genome assemblies. My feeling is that natural variation based on genome duplications remains plausible and compatible with the BUSCO data reported here. I suggest the authors to carry some new analyzes to understand the origin of the extra 150Mb that are present in the Gui et al. 2022 assembly : numbers of extra contigs, size, read coverage, composition (transposons, duplicated genes...) for example. The goal is to identify some peculiar properties of these extra contigs that would be useful to decipher if there are “real” contigs or assembly artifacts.

R2-1: Thank you for the comment.

You are correct that genome size variation was only a minor focus in the original study by Gui et al. However, this aspect led to the publication of a subsequent paper by Huang et al., and this issue is not minor anymore. We believe that it was important to examine the reliability of the genome size reported by Gui et al.

In fact, we clarified in the manuscript (L68) that the sizes of PacBio-based assemblies range from 380 Mb to 390 Mb, not 320 Mb to 390 Mb, as mentioned in your comment. This information actually highlights the consistency in assembly size when the same sequencing technology is applied across different samples.

Nevertheless, we understand your point that the 543 Mb genome assembly reported by Gui et al. could potentially result from natural segmental duplications, including duplicated BUSCO genes. To investigate this possibility, we identified the genomic locations of the BUSCO gene duplications specific to Gui et al.'s assembly by mapping them onto the NCBI reference genome to detect any particular contigs showing signs of segmental duplication. However, we found that the duplicated BUSCO genes are distributed across all 31 chromosomes, with no evidence of ‘extra’ contigs harboring clustered duplications. This result suggests that the observed inflation is not due to natural segmental duplications. This result is described in L100-L107, and these BUSCO genes are reported in Table S1.

2) Even the genome size estimation using k-mer lead to substantial variations that are systematically minored by the authors. For instance, in Fig 2A, genome size estimation with GenomeScope vary from 290 Mb to 450 Mb. Even if we excluded some extreme outliers, there is a significant spread within the 300Mb-400Mb range. Same patterns occurs in Fig. 3 with some huge genome size estimation variations. I agree with the authors that, in average, most samples fell close to the range of 400Mb but for reasons I ignore a fraction of the sample clearly deviate from the distribution and the authors failed to explain why. Maybe it’s artifactual in relation with sequencing bias or something like this, but again, it might reflect true genome size variation... Again, additional analyzes would be useful to understand why. I suggest the authors to try to assemble the read of these atypical samples and to analyzes the composition of the contigs, especially transposable elements that appears to be the main candidates for genome expansion. This would bring some valuable information to exclude the possibility that there are some degrees of natural genome variation.

R2-2: We absolutely understand that the estimated variation in genome sizes can appear quite large. Indeed, it should not be surprising to observe genome size variation within species, and the key point is how to interpret the observed ranges in genome sizes. Thanks to your comment, we realized that this issue was not properly addressed in the manuscript.

To evaluate the genome size variation reported by Gui et al., we compared their estimates of genome sizes with our new estimates based on Genomescope. We found that the variation estimated by Genomescope is much narrower than that reported by Gui et al. This result demonstrates that genome size estimates can be strongly influenced by the method. These results were added to Figure 2A and L152-L160. We believe that this result strengthens our conclusion. Thank you again, and we appreciate it.

In addition, in response to your comment regarding transposable element activity, we also showed that genome size variation could result from differences in both repeat sequences and unique sequences. This result has been added to Figure S2, L151-L152, and L225-L226.

3) I fully agree with the authors that reference genome assemblies of Spodoptera species are all close to 400 Mb (Fig 4) but if we enlarge the scope, there is huge genome size variation in Lepidoptera with some of them far exceeding the 500 Mb range (up to 800 Mb). The argument seems to me very week because it critically depends on the the phylogenetic range that were examined.

R2-3: We completely agree with you that lepidopteran species have huge genome size variations. Please note, however, that we have made it clear that we are addressing the genome sizes only for the Spodoptera genus across the whole manuscript (L19-L21, L84-L85, L180-L189, and L195-L196).

Minor points :

L48-57 / L168-176 : I think there is a misunderstanding of the pan-genome concept. The fact that there is a pangenome of 1.37 Gb found in 135 individuals that are not present in the reference genome assembly (400Mb) do not mean that a single individuals may have a genome of 400 Mb + 1.37 Gb = 1.77 Gb as mentioned by the authors. This simply mean that the 1.37 Gb of extra DNA may be randomly distributed among the 135 accessions. These 2 paragraphs need to be re-written.

R2-4: You are absolutely right. We revised the first sentence (lines 53-56) accordingly. We kept the second part unchanged, as it does not imply that the genome size could reach the sum of the reference and non-reference genomes.

L181-183 : The last sentence is offensive and usefulness. Remove it or replace it by a more neutral conclusion.

R2-5: Thank you for this comment. We rewrite the sentence (L204-L206).

Figure 1 : Size and resolution of this figure need some improvements.

R2-6: Thank you for this suggestion. We increased the figure size with increased resolution.

Data availability : it should be valuable to depose the raw data of the BUSCO and kmer distribution analyzes somewhere (Zenodo or something like this)

R2-7: We appreciate this suggestion. In line with our commitment to scientific transparency, we have made the data publicly available on Figshare (L236).

---

## [Decision Letter · Decision Letter 1]

15 Jul 2025

PONE-D-25-25103R1Is the extreme within-population genome size variation real in Spodoptera frugiperda?PLOS ONE

Dear Dr. NAM,

Thank you for submitting your manuscript to PLOS ONE. After careful consideration, we feel that it has merit but does not fully meet PLOS ONE’s publication criteria as it currently stands. Therefore, we invite you to submit a revised version of the manuscript that addresses the points raised during the review process.

We look forward to receiving your revised manuscript.

Kind regards,

Vivekanandhan Perumal, Ph.D

Academic Editor

PLOS ONE

Journal Requirements:

Additional Editor Comments:

Dear authors, thank you for responding to most of the reviewers' comments. However, some errors still remain. Please carefully revise the manuscript in accordance with the reviewers' feedback.

Reviewers' comments:

Reviewer's Responses to Questions

**Comments to the Author**

1. If the authors have adequately addressed your comments raised in a previous round of review and you feel that this manuscript is now acceptable for publication, you may indicate that here to bypass the “Comments to the Author” section, enter your conflict of interest statement in the “Confidential to Editor” section, and submit your "Accept" recommendation.

Reviewer #1: All comments have been addressed

Reviewer #2: All comments have been addressed

2. Is the manuscript technically sound, and do the data support the conclusions?

Reviewer #1: Yes

Reviewer #2: Yes

3. Has the statistical analysis been performed appropriately and rigorously? 

Reviewer #1: Yes

Reviewer #2: Yes

4. Have the authors made all data underlying the findings in their manuscript fully available?

Reviewer #1: Yes

Reviewer #2: Yes

5. Is the manuscript presented in an intelligible fashion and written in standard English?

Reviewer #1: Yes

Reviewer #2: Yes

6. Review Comments to the Author

Reviewer #1: Thank you for your careful revisions to address all the comments raised. After reviewing your responses and the corresponding modifications in the manuscript, I am satisfied with the improvements made.

Regarding the spelling and format issues, the unification of terms, clarification of abbreviations (e.g., "mya"), correction of table number references, and standardization of software name formatting (e.g., specifying "Genome Complexity Estimator (GCE)" for the first mention) have been properly handled, enhancing the consistency and readability of the text.

For the concerns about assembly strategies and genome size, while the exact cause of genome size inflation in Gui et al.’s work remains challenging to pinpoint due to limited data availability, the addition of assembly statistics in Table 1 and relevant text (L211–212) provides valuable context for readers to evaluate assembly quality, which is a reasonable approach within the scope of this study.

The clarification of sample sources (Zhang et al.’s samples mostly from maize fields, with two exceptions from sugarcane) in L213–216 addresses the representativeness issue of geographical population samples, providing necessary context for interpreting host-related analyses.

The inclusion of detailed methodological information and confidence intervals for divergence time estimation (L183–187) strengthens the robustness of the evolutionary stability analysis.

The supplementation of specific details from the original literature for flow cytometry data (L63–65) and the specification of software versions (e.g., Genomescope v1.0 in L224–225) improve the transparency and reproducibility of the research methods.

The addition of specific examples (100% protein sequence identity cases in Table S3, L128; metazoan BUSCO genes present in both reference and non-reference assemblies in L123–126 and L129) effectively supports the verification of non-reference sequence contamination, enhancing the persuasiveness of the related conclusions.

Overall, the revisions have adequately addressed all the raised concerns, and the manuscript now meets the requirements for publication. I recommend accepting the manuscript in its current form.

Reviewer #2: The authors have addressed convincingly my initial comments and I believe that the manuscript is now suitable for publication

7. PLOS authors have the option to publish the peer review history of their article (what does this mean? ). If published, this will include your full peer review and any attached files.

**Do you want your identity to be public for this peer review?** For information about this choice, including consent withdrawal, please see our Privacy Policy .

Reviewer #1: **Yes: ** Shaoying Wu

Reviewer #2: **Yes: ** Jonathan Filée

---

## [Author Response · Author response to Decision Letter 2]

19 Jul 2025

Reviewer #1: Thank you for your careful revisions to address all the comments raised. After reviewing your responses and the corresponding modifications in the manuscript, I am satisfied with the improvements made.

Regarding the spelling and format issues, the unification of terms, clarification of abbreviations (e.g., "mya"), correction of table number references, and standardization of software name formatting (e.g., specifying "Genome Complexity Estimator (GCE)" for the first mention) have been properly handled, enhancing the consistency and readability of the text.

For the concerns about assembly strategies and genome size, while the exact cause of genome size inflation in Gui et al.’s work remains challenging to pinpoint due to limited data availability, the addition of assembly statistics in Table 1 and relevant text (L211–212) provides valuable context for readers to evaluate assembly quality, which is a reasonable approach within the scope of this study.

The clarification of sample sources (Zhang et al.’s samples mostly from maize fields, with two exceptions from sugarcane) in L213–216 addresses the representativeness issue of geographical population samples, providing necessary context for interpreting host-related analyses.

The inclusion of detailed methodological information and confidence intervals for divergence time estimation (L183–187) strengthens the robustness of the evolutionary stability analysis.

The supplementation of specific details from the original literature for flow cytometry data (L63–65) and the specification of software versions (e.g., Genomescope v1.0 in L224–225) improve the transparency and reproducibility of the research methods.

The addition of specific examples (100% protein sequence identity cases in Table S3, L128; metazoan BUSCO genes present in both reference and non-reference assemblies in L123–126 and L129) effectively supports the verification of non-reference sequence contamination, enhancing the persuasiveness of the related conclusions.

Overall, the revisions have adequately addressed all the raised concerns, and the manuscript now meets the requirements for publication. I recommend accepting the manuscript in its current form.

R1: Thank you for recommending the publication of this manuscript without further revision.

Reviewer #2: The authors have addressed convincingly my initial comments and I believe that the manuscript is now suitable for publication

R2: Thank you for recommending the publication of this manuscript.

---

## [Decision Letter · Decision Letter 2]

3 Sep 2025

Is the extreme within-population genome size variation real in Spodoptera frugiperda?

PONE-D-25-25103R2

Dear Dr. NAM,

We’re pleased to inform you that your manuscript has been judged scientifically suitable for publication and will be formally accepted for publication once it meets all outstanding technical requirements.

Kind regards,

Vivekanandhan Perumal, Ph.D

Academic Editor

PLOS ONE

Additional Editor Comments (optional):

The authors have adequately addressed the comments from both reviewers. The revised manuscript is scientifically sound, and I recommend it for publication.

Reviewers' comments:

Reviewer's Responses to Questions

**Comments to the Author**

1. If the authors have adequately addressed your comments raised in a previous round of review and you feel that this manuscript is now acceptable for publication, you may indicate that here to bypass the “Comments to the Author” section, enter your conflict of interest statement in the “Confidential to Editor” section, and submit your "Accept" recommendation.

Reviewer #1: All comments have been addressed

2. Is the manuscript technically sound, and do the data support the conclusions?

Reviewer #1: Yes

3. Has the statistical analysis been performed appropriately and rigorously? 

Reviewer #1: Yes

4. Have the authors made all data underlying the findings in their manuscript fully available?

Reviewer #1: Yes

5. Is the manuscript presented in an intelligible fashion and written in standard English?

Reviewer #1: Yes

6. Review Comments to the Author

Reviewer #1: (No Response)

7. PLOS authors have the option to publish the peer review history of their article (what does this mean? ). If published, this will include your full peer review and any attached files.

**Do you want your identity to be public for this peer review?** For information about this choice, including consent withdrawal, please see our Privacy Policy .

Reviewer #1: No

---

## [Editor Report · Acceptance letter]

PONE-D-25-25103R2

PLOS ONE

Dear Dr. NAM,

I'm pleased to inform you that your manuscript has been deemed suitable for publication in PLOS ONE. Congratulations! Your manuscript is now being handed over to our production team.

Kind regards,

on behalf of

Dr. Vivekanandhan Perumal

Academic Editor

PLOS ONE